# Possible Toxic Mechanisms of Deoxynivalenol (DON) Exposure to Intestinal Barrier Damage and Dysbiosis of the Gut Microbiota in Laying Hens

**DOI:** 10.3390/toxins14100682

**Published:** 2022-09-30

**Authors:** Xiaohu Zhai, Zhi Qiu, Lihua Wang, Youwen Luo, Weihua He, Junhua Yang

**Affiliations:** 1Institute of Pet Science and Technology, Jiangsu Agri-Animal Husbandry Vocational College, Taizhou 225300, China; 2Institute for Agri-Food Standards and Testing Technology, Shanghai Academy of Agricultural Sciences, Shanghai 201403, China

**Keywords:** deoxynivalenol, laying hens, intestinal inflammation, barrier function, intestinal microbiota

## Abstract

Deoxynivalenol is one the of most common mycotoxins in cereals and grains and causes a serious health threat to poultry and farm animals. Our previous study found that DON decreased the production performance of laying hens. It has been reported that DON could exert significant toxic effects on the intestinal barrier and microbiota. However, whether the decline of laying performance is related to intestinal barrier damage, and the underlying mechanisms of DON induced intestine function injury remain largely unclear in laying hens. In this study, 80 Hy-line brown laying hens at 26 weeks were randomly divided into 0, 1, 5 and 10 mg/kg.bw (body weight) DON daily for 6 weeks. The morphology of the duodenum, the expression of inflammation factors and tight junction proteins, and the diversity and abundance of microbiota were analyzed in different levels of DON treated to laying hens. The results demonstrated that the mucosal detachment and reduction of the villi number were presented in different DON treated groups with a dose-effect manner. Additionally, the genes expression of pro-inflammatory factors IL-1β, IL-8, TNF-α and anti-inflammatory factors IL-10 were increased or decreased at 5 and 10 mg/kg.bw DON groups, respectively. The levels of ZO-1 and claudin-1 expression were significantly decreased in 5 and 10 mg/kg.bw DON groups. Moreover, the alpha diversity including Chao, ACE and Shannon indices were all reduced in DON treated groups. At the phylum level, *Firmicutes* and *Actinobacteria* and *Bacteroidetes*, *Proteobacteria*, and *Spirochaetes* were decreased and increased in 10 mg/kg.bw DON group, respectively. At the genus levels, the relative abundance of *Clostridium* and *Lactobacillus* in 5 and 10 mg/kg.bw DON groups, and *Alkanindiges* and *Spirochaeta* in the 10 mg/kg.bw DON were significantly decreased and increased, respectively. Moreover, there were significant correlation between the expression of tight junction proteins and the relative abundance of *Lactobacillus* and *Succinispira.* These results indicated that DON exposure to the laying hens can induce the inflammation and disrupt intestinal tight junctions, suggesting that DON can directly damage barrier function, which may be closely related to the dysbiosis of intestinal microbiota.

## 1. Introduction

Mycotoxins are a class of secondary metabolites produced by different fungal species that are distributed in a wide range of agricultural commodities worldwide, and they pose a great threat to human and animal health [1]. It is estimated that 25% of global crops are highly susceptible to mycotoxin contamination during growth, storage, and processing if the conditions are favorable for fungi growth [2]. Deoxynivalenol (DON), also called vomitoxin, is one kind of mono-terminal mycotoxins produced by filamentous fungi of the genus *Fusarium* under suitable conditions of temperature and humidity, mainly presented in wheat and corn for the grains and feeds. Due to its high stability and temperature resistance, DON still be detected in the wheat after four years of storage after contamination [3]. In the past three years, DON was reported to be one of the most contaminated fungal mycotoxins of feeds in China [4]. Accumulated data strongly indicate that DON exerts toxic effects on experimental animals, livestock, and humans that not only affect the growth performance and nutrient absorption of animals but also causes anorexia, vomiting, and diarrhea; long-term exposure to DON can lead to intestinal and immune dysfunction, shock or even death in livestock [4,5,6]. Annual economic loss of the livestock industries has been estimated as much as several hundred million dollars [7].

The layer breeding industry plays a pivotal role in the development of animal husbandry. Many recent evidences suggest that there is a close correlation between low-dose fungal toxins (DON) and chicken growth and health status. Previous information indicated that low doses of DON in the feed induced the increase of the subclinical necrotizing enteritis and negatively affected the intestinal barrier function, which suggested that the intestinal health was the key to maintain healthy production in poultry [8]. Accordingly, the intestine is the first barrier between the body and external substances, which not only prevents the invasion of exogenous heterogeneous substances such as feed ingredients/compounds, biotoxins and microorganisms but also easily becomes an attacked target by exogenous substances. Recent studies have focused on the damage effect of DON on the animal intestine, which found that DON not only interferes with the digestion and absorption of nutrients in the intestine, but also decreases the expression of tight junction proteins to disrupt the intestinal barrier function, causes intestinal mucosal damage and reduces the intestinal immune response in pigs and rodents [9,10]. Preliminary studies in our laboratory showed that DON induced the change of the diversity and species abundance of intestinal microbiota in mice, and intestinal damage was closely related to the changes in intestinal microbiota [11]. 

At present, there are limited studies on the toxic effects of DON contamination on laying hens, but our previous study found that DON treatment could reduce the laying performance and egg quality and cause damage to the liver and kidney tissue in laying hens. Additionally, a diet combinedly contaminated by aflatoxin B1 and DON significantly affected the egg production of laying hens, although there was a gap that the influence of DON on the laying performance was related to intestinal damage [12]. Mounting evidence about the birds suggested that DON exposure injured the morphological structure and integrity of intestinal tissues, downregulating the expression of tight junction proteins, reducing the transepithelial resistance, and increasing the permeability of intestinal epithelial cells; this in turn has often been accompanied with breaking the balance of intestinal microbiota and led to the reduced utilization of energy and nutrients and the decreased production performance of broilers [13,14,15,16,17]. These findings indicate that DON contamination in feed poses a serious threat to the health of poultry and causes huge economic losses to the farming industry.

Unfortunately, there are few reports on the study of DON involved in the intestinal damage of laying hens. Therefore, this study was conducted to explore the effects of DON with different concentrations on the morphological and tissue structure, intestinal barrier integrity, and inflammatory factors of laying hens. In addition, the effects of different DON concentrations on the structural composition, microbial diversity, and species abundance of the intestinal microbiota in laying hens were investigated using Illumina-MiSeq high-throughput sequencing technology. The aim of this study was to explore the underlying mechanisms of DON-induced intestine function injury in laying hens, and the relevance between the intestinal barrier damage and the imbalance of gut microbiota, which will provide a new reference for the research of mycotoxins induced intestinal toxicity, and the safety evaluation of mycotoxin contamination in the feed of laying hen.

## 2. Results

### 2.1. Effect of DON Treatment on the Intestinal Histopathology

The morphology of the duodenum in laying hens exposed to DON is shown in Figure 1. Compared with control, the intact intestinal structure was presented with neat villi and regular crypt morphology in the 1 mg/kg.bw DON group (Figure 1B). Additionally, the villus quantity and length presented the decreasing tendency in 5 and 10 mg/kg.bw DON groups (Figure 1C,D). Moreover, the villus in 10 mg/kg.bw DON group were seriously broken and dislodged (Figure 1D). 

### 2.2. Effect of DON Treatment on the Expression of Inflammatory Factors in the Duodenum of Laying Hens

Compared with the control, the birds that received 5 and 10 mg/kg.bw DON, showed a significant a dose-dependent increase in the expression of pro-inflammatory factor IL-1β, IL-8 and TNF-α (*p* < 0.05 or *p* < 0.01, Figure 2A–C), but the expression of IL-1β and TNF-α were only slightly increased in 1 mg/kg.bw DON group (*p* > 0.05). Accordingly, the anti-inflammatory factor IL-10 in the intestine of laying hens treated with 5 and 10 mg/kg.bw DON was significantly lower than that in the control group (*p* < 0.05 or *p* < 0.01, Figure 2D). However, the level of the anti-inflammatory factor slightly decreased at 1 mg/kg.bw DON group (*p* > 0.05).

### 2.3. Effect of DON Treatment on the Expression of Tight Junction Proteins in the Duodenum of Laying Hens

As shown in the Figure 3A_1_,3B_1_, the mRNA expression of intestinal tight junction ZO-1 and claudin-1 in 5 and 10 mg/kg.bw DON groups were significantly decreased when compared to the control group (*p* < 0.05). Similarly, the relative expression of ZO-1 and claudin-1 preteins were also markedly reduced in DON treated groups (*p* < 0.05 or *p* < 0.01), and there was a dose-dependent manner (Figure 3A_1_,3A_2_,3B_1_,3B_2_). However, the expression of ZO-1 and claudin-1 in 1 mg/kg.bw DON group was slightly decreased, and there was no significant difference between DON treated and control group (*p* > 0.05).

### 2.4. Effect of DON Treatment on Diversity of Fecal Microbiota in the Laying Hens

As shown in Table 1, there was no significant change in Simpson index between the DON-treated groups and the control group (*p* > 0.05). Compared with the control, the Chao and ACE indices were significantly decreased in the 1, 5 and 10 mg/kg.bw DON groups (*p* < 0.01), and the Chao and ACE indices in the 5 mg/kg.bw DON group were lower than that in 1 mg/kg.bw DON group (*p* < 0.05); in contrast, the indices in the 10 mg/kg.bw DON group were higher than those in the 5 mg/kg.bw DON group (*p* < 0.05). Similarly, the Shannon index in 5 and 10 mg/kg.bw DON groups were significantly lower than that in the control group (*p* < 0.01 or *p* < 0.05), but which was lower than that in 1 mg/kg.bw DON group (*p* < 0.01 or *p* < 0.05). And the Shannon index in 10 mg/kg.bw DON group was slightly higher than that in 5 mg/kg.bw DON group (*p* < 0.05).

### 2.5. Effect of DON Treatment on the Abudance of Intestinal microbiota in Laying Hens

At the phylum and genus levels, microbial abundance ratios in the top 10, higher than 0.1%, were selected for taxonomic annotation and difference analysis (Figure 4A,B). At the phylum level shown in Table 2, the analysis of the first six phyla showed that the relative abundances of Firmicutes and Actinobacteria in the DON-treated groups were reduced in a dose-dependent manner and that only in 10 mg/kg.bw DON group was significantly decreased compared with the control (*p* < 0.05). Contrarily, the relative abundances of Bacteroidetes, Proteobacteria, and Spirochaetes in DON treated group were increased with a dose-dependent relation, and there were all significant differences between 10 mg/kg.bw DON group and control group (*p* < 0.05). Moreover, the relative abundances of Fusobacteria in the DON-treated groups also increased, and there was only a significant difference in the 5 mg/kg.bw DON group compared with control (*p* < 0.05).

At the genus levels, the change of intestinal bacteria abundance in laying hens could be observed in Table 2. Compared with the control, the relative abundance of Bacteroides, Barnesiella, Succinispira, Prevotella, Helicobacter, and Euryarchaeota were slightly increased in all DON treatment groups with a dose-dependent relation, but there was no significant difference (*p* > 0.05). In addition, the relative abundance of Clostridium and Lactobacillus in 5 and 10 mg/kg.bw DON groups were all lower than that in control group (*p* < 0.05), but there was no significant difference between the 1 mg/kg.bw DON group and the control group (*p* > 0.05). Moreover, the relative abundance of Alkanindiges and Spirochaeta were significantly increased in the 10 mg/kg.bw DON group compared with the control (*p* < 0.05), and there were slightly increase in 1 and 5 mg/kg.bw DON groups compared to the control (*p* > 0.05).

### 2.6. Effect of DON Treatment on the Correlations between Tight Junction Proteins and Intestinal Microbiota in Laying Hens

To further understand the role of the intestinal microbiota in regulating intestinal barrier function, their relationship was analyzed by Spearman’s correlation analysis. The results are shown in Figure 5, the top 10 microbial genera were significantly correlated with the mRNA and proteins expression of ZO-1 and cloudin-1. Among the microbial genera downregulated by DON, *Lactobacillus* had a positive correlation (*p* < 0.01) with ZO-1 and cloudin-1 levels. However, the *Succinispira* presented a negative correlation with the expression of ZO-1 and cloudin-1 (*p* < 0.01).

## 3. Discussion

In this present study, we investigated the possible causes of intestinal function damage after DON exposure. We examined the intestinal inflammation, barrier function and changes of intestinal microbiota in laying hens exposed to DON. The results showed that DON damaged the histomorphological structure and disrupted the intestinal integrity of the intestine, induced the inflammation response and broke the balance of the intestinal microbe in laying hens. These findings will provide an essential reference to explore the relationship between intestinal microbe and intestinal barrier function damage after DON exposure in the laying hens.

Our previous studies found that DON exposed to laying hens resulted in the decrease of egg laying rate [18], which was suggested to be mainly associated with the obstruction of digestion and absorption of nutrients [19]. The intestines were the important digestive organs, and the damage to the intestinal function may lead to poor absorption of nutrients and inadequate supply of nutrients to the organism. Our founding showed that DON exposure led to the decrease of the small intestinal villi. A similar study indicated that DON was able to significantly affect the histomorphological structure of small intestine in the broiler, causing a decrease of villi height and an increase of crypt depth [20]. Another information found that the intestinal villi based on culturing the jejunum tissue of pig in vitro gradually appeared to be flat and fused treated by DON, and were accompanied by focal loss of apical enterocytes, villi atrophy and apical exfoliation [21]. These similar pathological changes confirmed that DON exposure affected intestinal nutrient digestion and absorption, which may be closely related to intestinal villi detachment and morphological damage, and thus destroying intestinal barrier function.

Intestinal barrier damage is an important result of intestinal dysfunction. Under normal physiological conditions, the intestinal mucosa can effectively prevent harmful microorganisms or various toxic and harmful substances from entering the organism through the intestine. Our findings revealed that DON exposure decreased the expression of the tight junction proteins ZO-1 and claudin 1, which was very likely to cause intestinal barrier damage. These results were consistent with previous findings that the expression of claudin-3, ZO-1, and occludins were decreased in the intestines of BALB/c mice and pigs after exposure to DON [22]. At the cellular level, the expression of ZO-1 was significantly reduced after DON treatment in IPEC-1 and IPEC-J2 cells for 48 h [23]. These similar results demonstrated that DON downregulated the intestinal tight junction proteins expression and disrupted the barrier function and integrity of the intestinal epithelium, which was in line with intestinal villi detachment and morphological damage. Therefore, the damage of tight junction integrity not only increased intestinal paracellular permeability, but also caused the increase of intestinal inflammation response and bacterial translocation, which will take an impact on the balance of the intestinal microbiota [9].

Inflammation is initiated as a protective response by the host, and could often result in a systemic toxicity. Our previous results showed that the pathological injury was caused by exposure to DON, which may be involved in the inflammatory response in order to maintain and regulate the integrity of the intestinal function. In the present study, the mRNA levels of IL-1β, IL-8, and TNF-α were significantly increased at 5 and 10 mg/kg.bw DON groups compared with the control, indicating that DON at this dose range induced the inflammatory response in the intestine of laying hens. Similarly, DON exposed to the piglets stimulated the expression increase of the inflammatory factors IL-1β, TNF-α and IL-8 in the intestinal tissue [24]. Additionally, it was found that the activities of IL-8 and TNF-α, and the relative mRNA expression of IL-1β and IL-8 were all increased in IPEC-J2 after DON exposure [25]. Moreover, IL-10 is the most potent anti-inflammatory factor, which could decrease the inflammation-mediated organs injury [26,27]. Several evidence suggest that DON downregulated the mRNA levels of anti-inflammatory factor IL-10 in the fish and porcine intestine, which was in line with the results of the laying hens [28,29]. These findings suggested that DON-induced inflammatory injury of the intestine contributed to the release and recruitment of pro-inflammatory mediators and restrain anti-inflammatory factors, which promoted an inflammatory response and induced the intestinal barrier injury.

The homeostasis of the intestinal microbiota has an important impact on maintaining the growth and development, nutrient digestion and absorption, immune antagonism and other physiological functions of the body. Numerous studies have shown a high correlation between intestinal microbiota, intestinal barrier function and inflammatory response [30,31]. Intestinal microbial diversity was very important to maintain the health of the organism, and it was generally accepted that the high species diversity might reflect a more stable microbiota system, and prevent the pathogen colonization and enhance immunity [32]. In this study, DON exposure significantly reduced the alpha diversity of the intestinal microbiota of laying hens. These findings were in accordance with the reports that DON was added to the rabbits [33], the alpha diversity based on ACE, Shannon, Chao, and Simpson indexes were clearly decreased in the ileum and particularly in the caecum. Similarly, the Chao and Shannon indexes in the colon of the pigs were decreased after diet contaminated with DON [34]. All these studies have proved that DON might alter the diversity and richness of the intestinal microbiota, and the variation of the microbiome was clearly related to the development of disease [35].

Intestinal microbial abundance was clearly related to the intestinal function and immunomodulatory of the organism. At the phylum level, the abundance of pathogenic bacteria such as *Proteobacteria* and *Spirochaetes* were increased in the 10 mg/kg.bw DON group. It has been reported that *Proteobacteria* was closely associated with dysbiosis of the intestinal microbiota, and observed with the low abundance in healthy individuals but high abundance in patients with intestinal diseases, which implied that *Proteobacteria* may be an important marker of intestinal diseases [36]. Many recent evidences have suggested that *Proteobacteria* might be involved in intestinal microbial translocation and damage in HIV-infected patients, and the high abundance of which was found to cause hypoimmunity in the body [37]. *Spirochaetes*, as the potential pathogens of humans and other species, contained a variety of pathogenic bacteria including *Brachyspira hyodysenteriae*, which can cause widespread and severe mucous hemorrhagic colitis in growing pigs [38]. These evidences suggested that DON exposure induced the increase of pathogenic bacteria at the phylum level. Moreover, DON exposure increased the abundance of pathogenic bacteria such as *Helicobacte, Prevotella* and *Spirochaeta* at the genus level. It was indicated that the high abundance of *Helicobacte* colonization was found to be related with the mucosal permeability increase of the stomach and intestine, which could damage the function of the intestinal barrier and increase the susceptibility to external stimuli [39]. These results suggested that the intestinal microbiota disbalance could promote the increase of pathogenic bacteria, disrupt the intestinal physicochemical barrier, and then induce injury of immune function. Besides to increasing the microbial abundance of pathogenic bacteria, DON also influenced the abundance of some beneficial bacteria. At the phylum level, *Firmicutes* and *Bacteroidetes* were the two most component of microorganisms, which participated in the energy metabolism, storage and maintenance of intestinal mucosal functions of host [40]. Studies have revealed that the change in the ratio of *Firmicutes* to *Bacteroidetes* may be an important biomarker of intestinal microbial balance [41].

In this study, the abundance of *Firmicutes* was lower and that of *Bacteroidetes* was higher in all DON-treated groups compared with the control. This result indicated that DON exposed in the laying hens disrupted the balance of the intestinal microbiota and induced the decrease of intestinal absorption of nutrients, which was in line with the lower growth performance and feed conversion rate of the pig exposure to DON [5]. In addition, a significant decrease in the abundance of *Bacteroidetes* was also observed in intestinal inflammatory response and ulcers, suggesting that *Bacteroidetes* was closely associated with intestinal damage and reduced immunity [36]. At the genus level, *Lactobacillus*, as one of the common probiotic genera involved in lipid and lactose degradation, enhanced intestinal immune function and has a bearing on the intestinal barrier protection, and its corresponding additives have been developed to reduce the intestinal toxic effects induced by DON [42]. Some information also indicated that exposure to DON in basal diets caused the decrease in the relative abundance of *Lactobacillus*, indicating that the decreased number of beneficial intestinal bacteria was observed after DON exposure and raised the damage of intestine [42]. Our results showed that *Lactobacillus* presented a positive correlation with the expression of ZO-1 and claudin-1. Some studies indicated that *Lactobacillus* prevents intestinal epithelial barrier function from the inflammation injury, which may contribute to *Lactobacillus* and their exopolysaccharides (EPS) facilitated the STAT3 (signal transducer and activator of transcription 3) binding to the promoter of occludin and ZO-1 [43,44]. *Succinispira* as a succinate and amino acid utilizer can ferment succinate to provide a steady source of energy for the host [45]. In this study, *Succinispira* increased only slightly after DON exposure, which had a negative correlation with the expression of tight junction proteins. However, there was limited information about *Succinispira* in the intestine, and it may involve the intestinal barrier function, which needs further experimental research. These results showed that DON decreased beneficial bacteria such as *Firmicutes* and *Lactobacillus*, and led to disrupt immunity and barrier function. Moreover, the increased abundance of pathogenic bacteria such as *Proteobacteria* and *Helicobacte* might exacerbate intestinal damage and immune dysregulation after DON exposure.

## 4. Conclusions

These data clearly indicated that DON exposure might contribute to the damage of intestinal morphological structure, intestinal mucosal detachment and reduction in the number of villi. Underlying mechanisms of DON caused intestinal epithelial injury are that DON induced the expression increase of cellular inflammatory factors, and decreases the expression of the tight junction proteins and disrupt the barrier function and integrity of the intestine. Additionally, the diversity reduction and relative abundance change of the intestinal microbiota could be another available mechanism of intestinal barrier function injury induced by DON.

## 5. Materials and Methods

### 5.1. Chemicals

Deoxynivalenol powder (purity ≥ 99%) was purchased from Pribolab Pte. Ltd. (Qingdao, China) and dissolved in the distilled water. Total RNA extraction and reverse transcription kits, all real-time polymerase chain reaction (PCR) reagents, and the fecal DNA extraction kit were purchased from Tiangen Biochemical Technology Co., Ltd. (Beijing, China). All other chemicals, if not stated, were obtained from Sigma (Shanghai, China).

### 5.2. Experimental Design and Birds Feeding Management

To avoid unnecessary discomfort to birds, the experiments were conducted at the laying hens breeding laboratory of Shanghai Academy of Agricultural Sciences (SAAS), Shanghai, China. The protocol was approved according to the Guidelines of Animal Ethics Committee in SAAS (Ethical review number: SAASPZ0920001, provided on 25 June 2020).

26-week-old Hy-line brown laying hens with similar body weight (1.90 ± 0.2 kg), feed consumption, and consistent egg production rate were housed individual in wire-floored cages with a light cycle (16L:8D) and air-conditioned (20 ± 2 °C room temperature, and humidity: 50~60%) environment. The laying hens were fed with a commercial diet that meet or exceed the nutritional requirement standard for laying hens (Table 3), which was purchased from a feed company located at Shanghai, China.

A total of 80 birds were randomly divided into four groups with 4 replicates and 5 birds per replicate. The four experimental groups were: 0, 1, 5 and 10 mg/kg.bw DON in feed, and all the BDE-209 dosages were chosen based on our preliminary tests, and reports of European Food Safety Authority (EFSA) and Chinese National Standards [18,46,47]. DON groups were intragastrical administered with 1.5 mL of DON with different concentrations (1, 5 and 10 mg/kg.bw) between 5 and 6 p.m. each day. Meanwhile, the birds in the control group received an equal volume of distilled water one daily for 42 days. Throughout the experimental period, the laying hens had free access to feed and fresh water at all times. Any clinical signs and deaths were recorded.

### 5.3. Sampling

At the 40th days of experiment, the papers were used to the sloping wire floors of the cages, the fresh fecal samples devoid of uncontrolled particles and feathers, were collected from each replicate in different groups at 7:00–9:00 am, mixed, and immediately stored at 4 °C with long-term storage at −80 °C until processing. For the intestinal collection, 12 birds were randomly selected and decapitated after egg laying from each group. The fresh central portions of duodenum were taken within 10 min after death. Samples for RNA isolation and western blotting were rinsed with ice-cold phosphate buffered saline (PBS). Intestinal epithelia were separated manually from the underlying submucosal layer of the duodenum using the glass slides on ice. All samples were immediately frozen in liquid nitrogen. In addition, another portion of duodenal tissues for morphology was fixed in 10% neutral formaldehyde (pH 7.4) and stored at 4 °C for 24 h.

### 5.4. Histopathology 

After being fixed for 24 h, the intestinal samples were dehydrated through a serial alcohol gradient (70, 80, 90, and 100%), cleared in xylene and then embedded in paraffin wax. 5 μm sections were stained with haematoxylin and eosin (H&E) according to the histopathological routine techniques. Finally, these slides were scanned with a Panoramic Desk Histological Blade Scanner (3DHISTECH Ltd., Budapest, Hungary) and the images were captured using the panoramic viewer software (3DHISTECH Ltd., Budapest, Hungary).

### 5.5. Real-Time Quantitative PCR (RT-qPCR)

Based on the results of histological examination, the genes expression of the inflammatory factors and intestinal tight junction proteins were determined by RT-qPCR. Total RNA was extracted using the trizol one-step method (Tiangen, China). RNA concentration and purity were determined by NanoDrop One Microvolume UV-Vis Spectrophotometer (Thermo Fisher Scientific, Waltham, MA, USA). And the reverse transcription of RNA to cDNA was prepared using TransScript One-Step cDNA Removal and cDNA Synthesis SuperMix kit (Tiangen, Beijing, China).

The mRNA levels of inflammatory factors genes (IL-1β, IL-8, IL-10, TNF-α) and intestinal tight junction protein-related genes (ZO-1, claudin-1) were quantified using ABI-prism 7500 Sequence Detection System (Applied Biosystems, Inc., Foster City, CA, USA) and normalized to β-actin reference gene. Real-time quantitative PCR reaction was carried out in 20 μL reaction mixtures, containing 2 μL cDNA, 10 μL Transtart Top Green qPCR SuperMix (2×), 0.8 μL primers, 7.2 μL RNase-free H_2_O. In addition, real-time PCR amplification conditions was as follow: 94 °C for 30 s; 94 °C for 5 s, 55~60 °C for 30 s, 40 cycles. The 2^−ΔΔCt^ method was used to determine the relative expression of each gene compared to a reference gene. All samples were amplified in a single PCR run, and each amplification was repeated at least three times.

As shown in Table 4, the primers sequences for IL-1β, IL-8, IL-10, TNF-α, ZO-1, claudin-1 and β-actin were designed using Prime Premier 5.0 according to the chicken gene sequences in GenBank, and then synthesized by Sangon Biotech (Shanghai) Co., Ltd. (Shanghai, China).

### 5.6. Protein Extraction and Western Blotting

Quick-frozen intestinal tissues samples (∼0.1 g) were mechanically homogenized by a tissue grinder, after thawing in ice-cold Protein Ext Mammalian Protein Extraction Kit (Transgen Biotech, Beijing, China). After 30 min incubation on ice, the homogenate was centrifuged at 12 000 g for 10 min, and the supernatants were collected for determination of protein concentration using bovine serum albumin (BSA) kit.

40 μg proteins for each sample were separated in 10% sodium dodecyl sulfate polyacrylamide gel electrophoresis (SDS-PAGE). Then the separated proteins were transferred onto the polyvinylidene fluoride (PVDF) membranes, and then the membranes were completely blocked by 5% skim milk powder in TBST for 1 h at room temperature. Membranes were probed overnight (4 °C) with either anti-claudin-1 (51-9000; Thermo Fisher, Shanghai, China; 1:500), anti-ZO-1 (40-2300; Thermo Fisher, Shanghai, China; 1:1000), and GAPDH antibody (MA1-16757; Thermo Fisher, Shanghai, China; 1:3000). After washing with TBST, the blots were incubated with HRP-conjugated goat anti-rabbit IgG (Immunology Consultants Laboratory, Inc., Portland, OR, USA; 1:20,000) or HRP-conjugated goat anti-mouse IgG (Immunology Consultants Laboratory, Inc. Portland, OR, USA; 1:20,000) for 2 h at room temperature. Finally, the signals were captured and the intensities of proteins on the blots were quantified using ImageJ Software (National Institute of Health, Bethesda, MD, USA).

### 5.7. 16S rRNA High-Throughput Sequencing

0.2 g sample of fresh feces were aseptically collected from different groups, and the total DNA was extracted using the rapid DNA extraction kit using manufacturer’s instructions (Transgen Biotech, Beijing, China), and the quality and quantity of DNAs were measured with a NanoDrop spectrophotometer (ND-1000; Thermo Fisher Scientific, Waltham, MA, USA). Polymerase chain reaction (PCR) amplification of bacterial 16S rDNA V3-V4 region was performed using primers (forward primer: 5′-ACTCCTACGGG AGGCAGCA-3′, and reverse primer: 5′-GGACTACHVGGGTWTCTAAT-3′). Moreover, the PCR reaction system was 25 μL, including 2.5 μL buffer (10×), 2 μL dNTP, 1 μL DNA template, 1 μL forward primer, 1 μL reverse primer, 0.125 μL DNA synthase, 17.375 μL ddH_2_O. And the PCR amplification procedure was as follow: 94 °C for 5 min; 94 °C for 30 s, 50 °C for 30 s, 72 °C for 30 s, 30 cycles, finally at 72 °C for 7 min, and storaged at 4 °C. PCR amplification products were detected by 2% agarose gel electrophoresis, and then quantified by QuantiFluorTM-ST blue fluorescence quantification system, and constructed and sequenced using an Illumina MiSeq platform (http://www.personalbio.cn/, Shanghai, China; accessed on 16 March 2019). The bioinformatics of high quality sequences were analyzed in terms of: (1) operational taxonomic unit (OTU) classification, the valid sequences obtained after quality control were clustered at 97% sequence similarity using QIIME software by calling the UCLUS sequence comparison tool to obtain operable classification units OTU representative sequences; (2) performing substance classification, the OTUs were compared with the template sequences from the GreenGene database (http://greengenes.lbl.gov/cgi-bin/nph-index.cgi; accessed on 20 March 2021) to obtain the taxonomic information corresponding to each OUT; (3) alpha diversity analysis, based on the OTU information, the QIIME software alpha-diversity.py command was called to analyze the alpha diversity indices of Chao, ACE, Shannon and Simpson of the samples; (4) cluster analysis of species composition and relative abundance of samples at the phylum and genus levels to screen for fecal microflora with significant differences.

### 5.8. Data Preprocessing and Statistical Analysis 

All the statistical analyses were performed using SPSS 22.0 software (version 22.0, IBM, Chicago, IL, USA). The values for mRNA expression and protein levels were presented as fold-change relative to control group. Statistical analyses were performed by one-way analysis of variance (ANOVA), followed by Tukey post-hoc test. All of the data were expressed as mean ± standard deviation (SD). *p* ≤ 0.05 and *p* ≤ 0.01 was considered statistically significant or highly significant, respectively. 

## Figures and Tables

**Figure 1 toxins-14-00682-f001:**
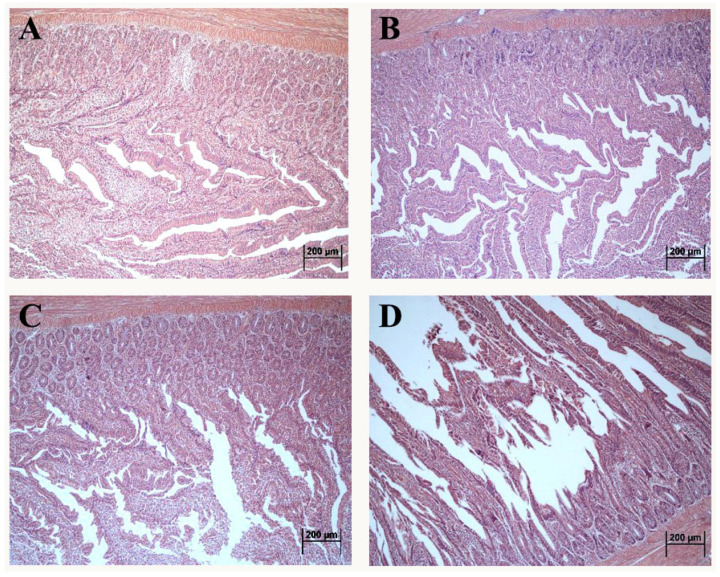
Altered morphological structure of duodenum in laying hens exposed to DON with different doses. (**A**) control group; (**B**) 1 mg/kg.bw DON group; (**C**) 5 mg/kg.bw DON group; (**D**) 10 mg/kg.bw DON group.

**Figure 2 toxins-14-00682-f002:**
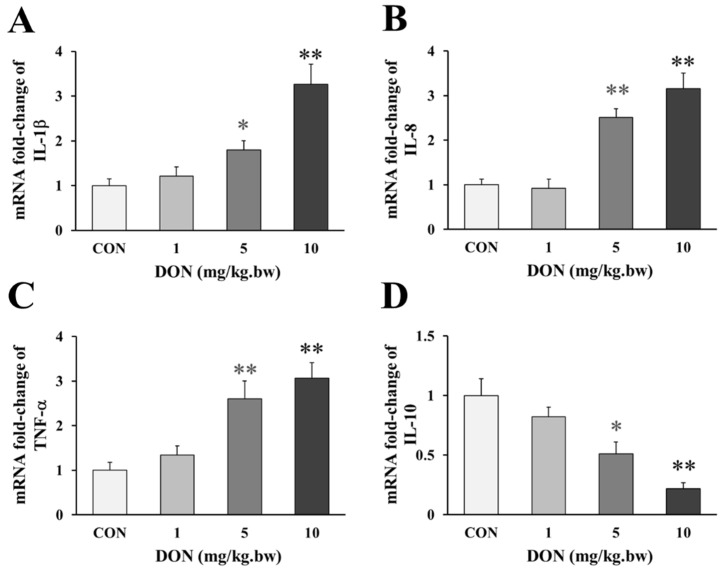
Change in genes expression of inflammatory factors in the duodenum of laying hens exposed to DON with different doses. (**A**) Relative expression of IL-1β; (**B**) Relative expression of IL-8; (**C**) Relative expression of TNF-α; (**D**) Relative expression of IL-10. β-actin was used as a reference gene. All the data are expressed as mean ± SD (*n* = 12). “*” and “**” mean significant differences (*p* < 0.05) or extremely significant difference (*p* < 0.01) compared with control group, respectively.

**Figure 3 toxins-14-00682-f003:**
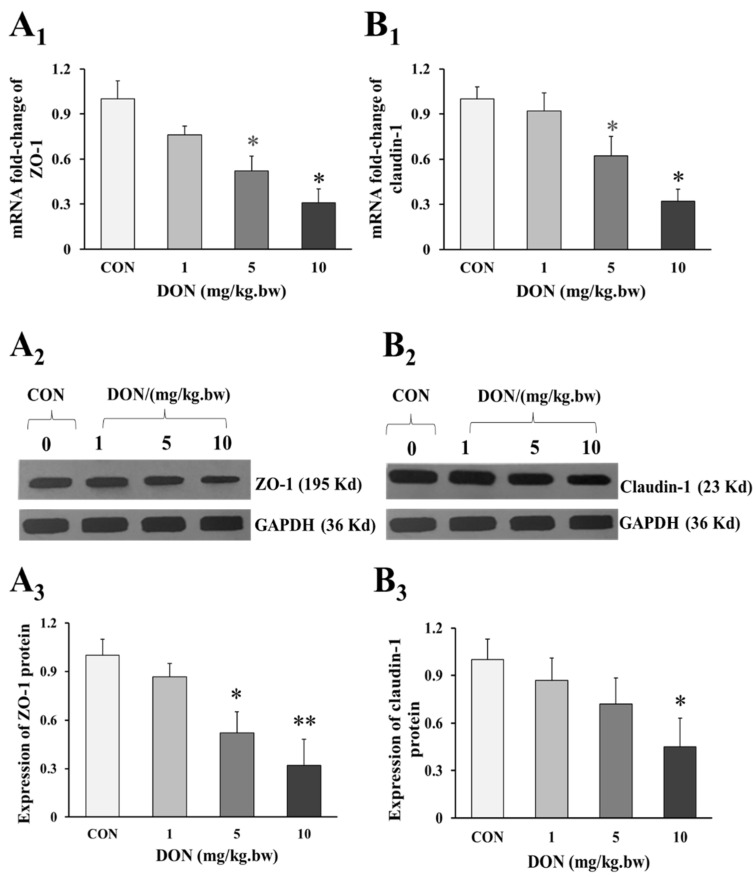
Change in expression of tight junction proteins in the duodenum of laying hens exposed to DON with different doses. (**A_1_**) mRNA relative expression of ZO-1; (**B_1_**) mRNA relative expression of claudin-1; (**A_2_**) Western blotting analysis of the expression of ZO-1; (**B_2_**) Western blotting analysis of the expression of claudin-1; (**A_3_**) Relative expression of ZO-1; (**B_3_**) Relative expression of claudin-1. β-actin or GAPDH was used as a reference gene or reference protein, respectively. All data are presented with mean ± SD (*n* = 12). “*” and “**” mean significant differences (*p* < 0.05) or extremely significant difference (*p* < 0.01) compared with control group, respectively.

**Figure 4 toxins-14-00682-f004:**
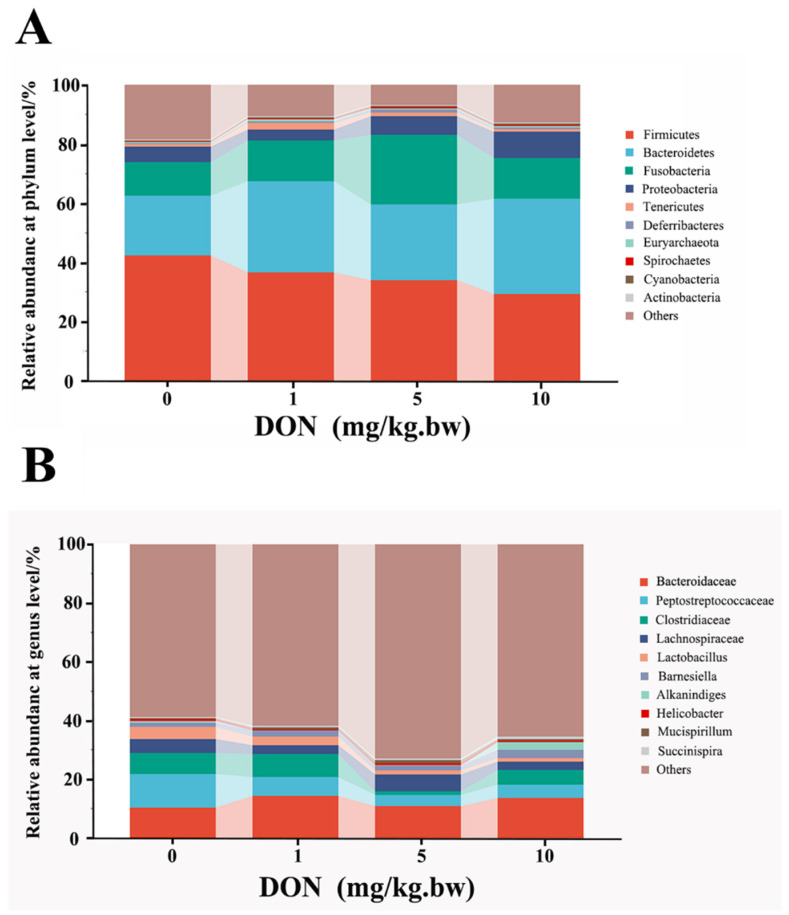
Change in phylum and genus abundance of intestinal microbes in the laying hens exposed to DON with different doses. The histogram shows the relative contents of major bacterial genera (top 10) at the phylum (**A**) and genus (**B**) levels in the intestinal microbiota of different groups. The data conformed to a normal distribution, and one-way analysis of variance was used (*n* = 4).

**Figure 5 toxins-14-00682-f005:**
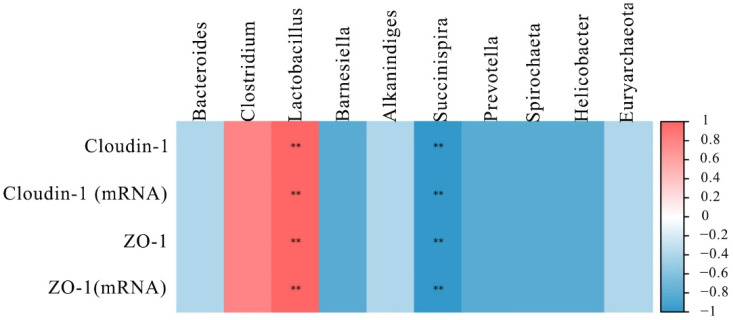
Heat map of correlation analysis between the expression of tight junction proteins and the microbial abundance in the intestine of laying hens. The intensity of the colors represented the degree of association (blue, negative correlation; red, positive correlation). “**” mean extremely significant correlations (*p* < 0.01).

**Table 1 toxins-14-00682-t001:** Change in Alpha diversity of intestinal microbiota in the laying hens exposed to DON with different doses.

Items	DON Concentration/(mg/kg.bw)			
	0	1	5	10
Chao index	1805.16 ± 28.30 ^Bd^	1545.95 ± 17.18 ^Abc^	1391.13 ± 66.77 ^Aa^	1584.86 ± 43.84 ^Ac^
ACE index	1800.42 ± 12.86 ^Cc^	1561.65 ± 22.71 ^ABb^	1418.78 ± 60.52 ^ABa^	1608.07 ± 33.70 ^Bb^
Simpson index	0.9567 ± 0.0051	0.9587 ± 0.0183	0.9507 ± 0.0298	0.9376 ± 0.0147
Shannon index	6.55 ± 0.24 ^Bc^	6.80 ± 0.29 ^Bc^	4.30 ± 0.72 ^Aa^	5.55 ± 0.40 ^ABb^

Note: All data are expressed as mean ± SD (*n* = 4). The different lower ease letters and capital letters in same row mean significant difference (*p* < 0.05) or extremely significant difference among different groups (*p* < 0.01), respectively.

**Table 2 toxins-14-00682-t002:** Change in abundance of intestinal microbes in the laying hens exposed to DON with different doses.

Target Microbes	DON Concentration/(mg/kg.bw)			
	0	1	5	10
**Phylum**				
*Firmicutes*	0.4237 ± 0.0344 ^b^	0.3405 ± 0.1147 ^ab^	0.2768 ± 0.094 ^ab^	0.2253 ± 0.0527 ^a^
*Bacteroidetes*	0.1642 ± 0.0377 ^a^	0.283 ± 0.0484 ^ab^	0.2562 ± 0.0412 ^ab^	0.3235 ± 0.1088 ^b^
*Fusobacteria*	0.1126 ± 0.0608 ^a^	0.138 ± 0.1035 ^ab^	0.2753 ± 0.07 ^b^	0.1592 ± 0.048 ^ab^
*Proteobacteria*	0.0538 ± 0.0243 ^a^	0.0369 ± 0.0135 ^ab^	0.0632 ± 0.0091 ^ab^	0.1012 ± 0.0302 ^b^
*Spirochaeta*	0.0019 ± 0.0017 ^a^	0.0037 ± 0.0027 ^ab^	0.0035 ± 0.0034 ^ab^	0.008 ± 0.0018 ^b^
*Actinobacteria*	0.0043 ± 0.0008 ^b^	0.0036 ± 0.0009 ^ab^	0.0031 ± 0.0009 ^ab^	0.0025 ± 0.0002 ^a^
**Genus**				
*Bacteroides*	0.1026 ± 0.0678	0.1415 ± 0.0772	0.107 ± 0.0979	0.1345 ± 0.0921
*Clostridium*	0.1144 ± 0.0129 ^b^	0.0659 ± 0.0402 ^ab^	0.0395 ± 0.0293 ^a^	0.0466 ± 0.0041 ^a^
*Lactobacillus*	0.0412 ± 0.0167 ^b^	0.0294 ± 0.0156 ^ab^	0.0133 ± 0.0055 ^a^	0.0094 ± 0.005 ^a^
*Barnesiella*	0.0148 ± 0.0103	0.0202 ± 0.0115	0.0164 ± 0.0157	0.0274 ± 0.0148
*Alkanindiges*	0.0046 ± 0.0076 ^a^	0.0005 ± 0.0001 ^a^	0.0014 ± 0.002 ^a^	0.0263 ± 0.0157 ^b^
*Succinispira*	0.0039 ± 0.0027	0.0057 ± 0.0032	0.0061 ± 0.0027	0.0065 ± 0.0037
*Prevotella*	0.0036 ± 0.0022	0.0057 ± 0.0031	0.0051 ± 0.0023	0.0071 ± 0.0041
*Spirochaeta*	0.0019 ± 0.0005 ^a^	0.0037 ± 0.0011 ^ab^	0.003 ± 0.0022 ^ab^	0.0058 ± 0.001 ^b^
*Helicobacter*	0.0065 ± 0.004	0.0041 ± 0.0054	0.0074 ± 0.0058	0.0183 ± 0.0257
*Euryarchaeota*	0.0029 ± 0.0018	0.0061 ± 0.0036	0.0041 ± 0.0043	0.0056 ± 0.0056

Note: All data are expressed as mean ± SD (*n* = 4). The different lower ease letters in same row mean significant difference among different groups (*p* < 0.05).

**Table 3 toxins-14-00682-t003:** Composition and nutrient levels of basal diets (air-dry basis).

Ingredients	Level/%	Calculated Nutrient Values	Level/%
Corn	61.3	Metabolizable energy ^(2)^	12.26
Wheat bran	3.0	Crude protein	19.79
Shell powder	4.0	Crude fiber	6.43
Soybean meal	23.8	Ether extract	3.54
Soybean oil	1.40	Ca	2.27
NaCl	0.5	P	0.56
Limestone	4.0	Lysine	0.76
Premix ^(1)^	2.0	Methionine	0.38
Total	100.00		

Note: ^(1)^ The premix provided the following per kg of the diet: VA 10 000 IU, VB_1_ 1.6 mg, VB_2_ 4.5 mg, VB_5_ 40 mg, VB_6_ 5 mg, VB_12_ 0.02 mg, VD_3_ 3 600 IU, VE 40 IU, VK_3_ 2 mg, Biotin 0.1 mg, folic acid 2.0 mg, D-pantothenic acid 10 mg, Cu 10 mg, Fe 35 mg, Mn 60 mg, Zn 50 mg, I 1.0 mg, Se 0.30 mg, Co 0.15 mg. ^(2)^ The unit of metabolizable energy is MJ/kg.

**Table 4 toxins-14-00682-t004:** Primer sequences for RT-qPCR.

Target Genes	GenBank Accession Number	Primer Sequences (5′–3′)	Product Length/bp	Annealing Temperature/°C
IL-1β	NM_204524.1	F: TACGAGATGGAAACCAGCAAC	84	58
		R: GGTCAACATCGCCACCTACAA		
IL-8	NM_205498.1	F: CATCTTTACCAGCGTCCTACC	106	55
		R: GAAAACAAGCCAAACACTCCT		
IL-10	EF554720	F: TAAGGACTATTTTCAATCCAGGG	142	55
		R: ACGGGGCAGGACCTCATC		
TNF-α	NM_204267.1	F: ACAGGGTAGGGGTGAGGATAG	184	55
		R: TGGGAGTGGGCTTTAAGAAGA		
ZO-1	XM_413773.4	F: TATTCTGAGGTGGAGGAGGGT	217	55
		R: TCTAAGGGGAAGCCAACTGAT		
claudin-1	NM_001013611.2	F: TCTCCAAATGCTTCTACTACCA	122	55
		R: AGTGAAACATCCTACCCACCC		
β-actin	NM_205518	F: TGCGTGACATCAAGGAGAAG	300	60
		R: TGCCAGGGTACATTGTGGTA		

## Data Availability

Data will be provided on request.

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
