# Peer review of "Possible Toxic Mechanisms of Deoxynivalenol (DON) Exposure to Intestinal Barrier Damage and Dysbiosis of the Gut Microbiota in Laying Hens"

_toxins, 2022, doi:10.3390/toxins14100682_

Round 1

Reviewer 1 Report

The type of paper should be changed to "Communication".

The abstract should be better summarized.

The context should be better described.

The novelty character of paper should be marked in the aim.

Data in figures 2 and 3 should be better described in the text.

Results in table 2 should be better described and discussed.

Author Response

Question 1: The type of paper should be changed to "Communication".

Response: Thanks very much for the reviewer’s good comments. We accept the suggestion that the type of paper will be changed into “Communication”.

Question 2: The abstract should be better summarized.

Response: Thanks very much for the reviewer’s good comments. We have made the corresponding supplement according to the reviewer's suggestions.

Question 3: The context should be better described.

Response: Thanks very much for good comments, we have rewritten this part.

Question 4: The novelty character of paper should be marked in the aim.

Response: Special thanks for reviewer’s comments, we have made the corresponding supplement according to the reviewer's suggestions.

Question 5:Data in figures 2 and 3 should be better described in the text.

Response: Special thanks for reviewer’s comments, we have rewritten this part according to the reviewer's suggestions.

Question 6:Results in table 2 should be better described and discussed.

Response: Thanks very much for good comments, we have rewritten this part according to the reviewer's suggestions.

Reviewer 2 Report

The study explore the effects of DON (Deoxynivalenol) with different concentrations on the morphological and tissue structure, intestinal barrier integrity, inflammatory factors, the effects on the composition, microflora diversity and species abundance of the intestinal microflora of laying hens. In this study, 80  laying hens at 26 weeks were randomly treated into 0, 1, 5 and 10 mg/kg DON daily for 6 weeks. Results indicated that DON exposure to the laying hens can induce the inflammation, disrupt intestinal tight junctions, and affect the diversity and species abundance of intestinal microflora, suggesting that DON can directly damage barrier function, which may be closely related to disorder of microflora in the intestine.

The article is well structured and organized. The methodologies are well explained as well as the presentation of the problem related to the exposure to DON and the results of the study.

Author Response

Reviewer 2#:

Question 1:The study explored the effects of DON (Deoxynivalenol) with different concentrations on the morphological and tissue structure, intestinal barrier integrity, inflammatory factors, the effects on the composition, microflora diversity and species abundance of the intestinal microflora of laying hens. In this study, 80 laying hens at 26 weeks were randomly treated into 0, 1, 5 and 10 mg/kg DON daily for 6 weeks. Results indicated that DON exposure to the laying hens can induce the inflammation, disrupt intestinal tight junctions, and affect the diversity and species abundance of intestinal microflora, suggesting that DON can directly damage barrier function, which may be closely related to disorder of microflora in the intestine.

The article is well structured and organized. The methodologies are well explained as well as the presentation of the problem related to the exposure to DON and the results of the study.

Response: Thanks a lot for reviewer’s affirmation and encouragement, we will continuously improve the quality of our paper.

Reviewer 3 Report

 I have read carefully the manuscript entitled “Possible Toxic Mechanisms of Deoxynivalenol (DON) Exposure to Intestinal Barrier Damage and Microflora Disorder in  Laying Hens” submitted to Toxins as an original article. In my opinion, Authors didn't draft the paper well and I cannot recommend the publication of the manuscript .

The main concerns are as follow:

 1.      Experimental layout: The experimental setup is completely unclear. The experimental groups were formed on the basis on DON content in kg of feed (L145) while hens were intragastrical administrated with DON ?? Throughout the manuscript, it is not possible to say whether DON was thus administered per kg feed or body weight. In the introduction (L89) Authors state that they aimed to provide additional reference for the safety evaluation of mycotoxin contamination in the feed, therefore I suppose that they idea was to establish the experimental doses on the feed level. This cannot be achieved by intragastrical administration of DON.

2.      Dose selection: how these doses (per kg of feed or bw)  correspond to doses at which birds are normally exposed ? How do they correspond to e.g.  EFSA or national regulations ??

3.      Feed: was feed (grain) checked for the presence of DON and other mycotoxins ?

4.      WB: catalog numbers of used antibodies are needed as reviewer wasn’t able to find any chickens-specific ZO-1 antibody in thermo fisher catalogue. More importantly, the only chicken-specific anty-caludin-1 antibody was  produced using rabbit as a host. Why anti-mouse IgG secondary antibody was used ?

5.      Blots: whole, non-cropped blot should be provided as supplementary material.

6.      Morphology: results of morphological analysis of the duodenums were not performed or are not presented. Only performed measurements with proper statistical analysis allow to draw any conclusions about quantitative changes in intestinal morphology.

7.      Statistics: The description of the experimental layout, does not allow to asses, what was the experimental unit for each trait, except for the fecal samples, in which case the cage was experimental unit (n-4). Therefore, Table 1 and 2: how n=6 was obtained from n=4 replicate cages ? (see footnote for fig 4). In other cases, each time the n number should be given. Also was normality of data checked? For example, RT-qPCR data rarely show normal distribution.

Technical comment:

Nomenclature: gene names should be written in italic, proteins - normal font. When writtting about genes, proper genes names (CLDN1, TJP1) not proteins which are coded by them (cluadin-1, zo-1) should be used.

 Minor comments:

 L52 Reference needed

L56 ref [8] is about broiler chickens, they do not lay eggs

L57-59 food ingredients/compounds and biotoxins are exogenous substances. please rephase.

L71-72 - this fragment can be removed, additional evidence about combined effects of DON+ALFA on eggshell quality does not provide any valuable information.

L67 “transepithelial resistance” usually correspond to TER, Transepithelial Electrical Resistance. That the Authors meant?

L83-84 Why none of these reports was presented in the introduction ?

L85 what tissue ?

L89 “provide additional reference”

Fig 2: markers of molecular weight are missing

Fig 4b- y-axis label: genus

L204 ecological?

L210 “intestines are”

L229 “occludins”

L343 Reference needed

L356 mixed between replicates or collection time points ?

L385 Was integrity of RNA checked ?

L387 Manufacturers ?

L389 cytokines instead of factors

L425 What kit exactly was used ?

L445 Chao1 or Chao2 ?

Author Response

The revision will be found in the appendix. 

Reviewer 4 Report

In lines 130 and 170, change the numbering of the figures

Line 236 the word “improvement”, I think better should be the word “increase”

Linie 248 The sentence about Il 10 at this point has no context with the sentence before and after it, isn't it better to delete this sentence or develop the thought about this interleukin?

Author Response

Reviewer 4

Question 1:In lines 130 and 170, change the numbering of the figures

Response: Thanks very much for good comments, we have made the corresponding correction according to reviewer’s suggestion.

Question 2:Line 236 the word “improvement”, I think better should be the word “increase”

Response: Special thanks for reviewer’s comments, we have rewritten it.

Question 3:Line 248 The sentence about Il 10 at this point has no context with the sentence before and after it, isn't it better to delete this sentence or develop the thought about this interleukin?

Response: Thanks very much for reviewer’s suggestion, we have deleted and written this sentence.

Round 2

Reviewer 3 Report

Q1. Satisfactory addressed

Q2. I meant to refer in your study recommendations from EFSA or other organizations regarding the amount of DON in the feed. These data can be easily found. For example, for EFSA (“Risks to human and animal health related to the presence of deoxynivalenol and its acetylated and modified forms in food and feed”doi.org/10.2903/j.efsa.2017.4718): “For laying hens, diets withDON concentrations up to 18 mg DON/kg feed did not induce any negative impact on body weightgain, hatchability and egg production in some studies. However, a diet of 10–13 mg DON/kg feedinduced a decrease of feed intake at an early stage of the experiment, and a decrease of relativeweights of spleen and gizzard and egg fertility. In other studies diets containing 5 mg DON/kg feed didnot affect body weight gain, hatchability and egg fertility. Therefore, the CONTAM Panel identified5 mg DON/kg feed as a NOAEL for laying hens.” For EU: Commision Recommendation 20016/1319 (https://eur-lex.europa.eu/legal-content/EN/TXT/PDF/?uri=CELEX:32016H1319&from=EN): guidance values for deoxynivalenol in animal feed: 8 mg/ kg feed (general cereals) and 12 mg/kg feed (maize by-products). FDA Advisory Levels for Deoxynivalenol (DON) in Finished Wheat Products for Human Consumption and Grains and Grain By-Products used for Animal Feed (https://www.fda.gov/regulatory-information/search-fda-guidance-documents/guidance-industry-and-fda-advisory-levels-deoxynivalenol-don-finished-wheat-products-human): “For chickens, 10 ppm DON on grains and grain by-products with the added recommendation that these ingredients not exceed 50% of the diet of chickens.”

Q3. Satisfactory addressed

Q4. Just an remark:  “the mRNA expression of ZO-1 and caludin-1 were used to verify the accuracy of WB”. The mRNA expression and protein immunolocalization are completely two different  phenomena and one cannot serve as verification for another. Just an example: Ghrelin-producing cells are most abundant in the stomach (mRNA expression), while ghrelin level can be assessed by WB or IHC in duodenum. The same for pancreatic hormones.

Q5. Form the Instruction for Authors from Toxins homepage: “In order to ensure the integrity and scientific validity of blots (including, but not limited to, Western blots) and the reporting of gel data, original, uncropped and unadjusted images should be uploaded as Supporting Information files at the time of initial submission. A single PDF file or a zip folder including all the original images reported in the main figure and supplemental figures should be prepared. Authors should annotate each original image, corresponding to the figure in the main article or supplementary materials, and label each lane or loading order. All experimental samples and controls used for one comparative analysis should be run on the same blot/gel image. For quantitative analyses, please provide the blots/gels for each independent biological replicate used in the analysis.” Also the information about molecular weight of analysed proteins should be added.

Q6. Without performance of proper statistical analysis the sentences like “Additionally, the decrease of villus quantity and length, and increase of crypt depth were observed in 5 and 10 mg/kg.bw DON treated groups” have no justification or grounds and should be removed.

Q7. No further comments.

Author Response

Question 1:Q1. Satisfactory addressed

Response: Thanks very much for the reviewer’s encouragement and assistance.

Question 1:Q2. I meant to refer in your study recommendations from EFSA or other organizations regarding the amount of DON in the feed. These data can be easily found. For example, for EFSA (“Risks to human and animal health related to the presence of deoxynivalenol and its acetylated and modified forms in food and feed”doi.org/10.2903/j.efsa.2017.4718): “For laying hens, diets with DON concentrations up to 18 mg DON/kg feed did not induce any negative impact on body weightgain, hatchability and egg production in some studies. However, a diet of 10–13 mg DON/kg feed induced a decrease of feed intake at an early stage of the experiment, and a decrease of relative weights of spleen and gizzard and egg fertility. In other studies diets containing 5 mg DON/kg feed did not affect body weight gain, hatchability and egg fertility. Therefore, the CONTAM Panel identified 5 mg DON/kg feed as a NOAEL for laying hens.” For EU: Commision Recommendation 20016/1319 (https://eur-lex.europa.eu/legal-content/EN/TXT/PDF/?uri=CELEX:32016H1319&from=EN): guidance values for deoxynivalenol in animal feed: 8 mg/ kg feed (general cereals) and 12 mg/kg feed (maize by-products). FDA Advisory Levels for Deoxynivalenol (DON) in Finished Wheat Products for Human Consumption and Grains and Grain By-Products used for Animal Feed (https://www.fda.gov/regulatory -information/search-fda-guidance-documents/guidance-industry-and-fda-advisory-levels-deoxynivalenol-don-finished-wheat-products-human): “For chickens, 10 ppm DON on grains and grain by-products with the added recommendation that these ingredients not exceed 50% of the diet of chickens.”

Response: Special thanks for reviewer’s comments, we have made the corresponding modifications of DON treated dose references according to the reviewer’ suggestion.

Question 3:Q3. Satisfactory addressed

Response: Thanks very much for the reviewer’s encouragement and assistance.

Question 4: Q4. Just an remark: “the mRNA expression of ZO-1 and caludin-1 were used to verify the accuracy of WB”. The mRNA expression and protein immunolocalization are completely two different phenomena and one cannot serve as verification for another. Just an example: Ghrelin-producing cells are most abundant in the stomach (mRNA expression), while ghrelin level can be assessed by WB or IHC in duodenum. The same for pancreatic hormones.

Response: We gratefully appreciate for your valuable suggestion. Generally, the mRNA expression of genes could be used to verify the accuracy of WB, but sometimes these two different phenomena can’t be completely verified each other. At present, the anti-ZO-1 for the chicken is very limited, but the molecular weight of protein in the gel is available and single, and the expression of protein is in line with the mRNA expression of genes. However, it is valuable to explore the suitable antibodies for chickens. Special thanks to the reviewer’s good comments again, which given us a good ideal in the future research.

Question 5: Q5. Form the Instruction for Authors from Toxins homepage: “In order to ensure the integrity and scientific validity of blots (including, but not limited to, Western blots) and the reporting of gel data, original, uncropped and unadjusted images should be uploaded as Supporting Information files at the time of initial submission. A single PDF file or a zip folder including all the original images reported in the main figure and supplemental figures should be prepared. Authors should annotate each original image, corresponding to the figure in the main article or supplementary materials, and label each lane or loading order. All experimental samples and controls used for one comparative analysis should be run on the same blot/gel image. For quantitative analyses, please provide the blots/gels for each independent biological replicate used in the analysis.” Also the information about molecular weight of analysed proteins should be added.

Response: Thanks a lot for reviewer’s good suggestion, and we have made the upload the WB blot/gel image as Supporting Information files. Thanks again for the reviewer’s good comments.

Question 6: Q6. Without performance of proper statistical analysis the sentences like “Additionally, the decrease of villus quantity and length, and increase of crypt depth were observed in 5 and 10 mg/kg.bw DON treated groups” have no justification or grounds and should be removed.

Response: Thanks very much for the good comments, and we have rewritten this part according to the reviewer’s suggestion.

Question 7: Q7. No further comments.

Response: Thanks very much for the reviewer’s encouragement and assistance.